# Proanthocyanidins Ameliorated Deficits of Lipid Metabolism in Type 2 Diabetes Mellitus Via Inhibiting Adipogenesis and Improving Mitochondrial Function

**DOI:** 10.3390/ijms21062029

**Published:** 2020-03-16

**Authors:** Fangfang Tie, Jifei Wang, Yuexin Liang, Shujun Zhu, Zhenhua Wang, Gang Li, Honglun Wang

**Affiliations:** 1Key Laboratory of Tibetan Medicine Research, Northwest Institute of Plateau Biology, Chinese Academy of Science, Xining 810008, China; fftie@nwipb.cas.cn (F.T.); wangjifei19@mails.ucas.ac.cn (J.W.); skywzh@ytu.edu.cn (Z.W.); 2Institutes of Life Science, University of Chinese Academy of Science, Beijing 100049, China; 3Center for Mitochondrial and Healthy Aging, College of Life Science, Yantai University, Yantai 264005, China; lemon96205@gamil.com (Y.L.); zhushujun2020@gmail.com (S.Z.)

**Keywords:** proanthocyanidins, flavan-3-ols, lipid metabolism, T2DM, 3T3-L1 adipocytes, mitochondrial function

## Abstract

Proanthocyanidins are the major active compounds extracted from *Iris lactea* Pall. var. *Chinensis* (Fisch.) Koidz (*I. lactea*). Proanthocyanidins exhibit a variety of pharmacological activities such as anti-oxidation, anti-inflammation, anti-tumor, and lowering blood lipids. However, the underlying mechanism of its regulating effect on lipid metabolism in diabetic conditions remains unclear. The present study investigated the effects of *I. lactea*-derived proanthocyanidins on lipid metabolism in mice of type 2 diabetes mellitus (T2DM). Results demonstrated a beneficial effect of total proanthocyanidins on dysregulated lipid metabolism and hepatic steatosis in high-fat-diet/streptozocin (STZ)-induced T2DM. To identify the mechanisms, six flavan-3-ols were isolated from proanthocyanidins of *I. lacteal* and their effects on adipogenesis and dexamethasone (Dex)-induced mitochondrial dysfunctions in 3T3-L1 adipocytes were determined. In vitro studies showed flavan-3-ols inhibited adipogenesis and restored mitochondrial function after Dex-induced insulin resistance, being suggested by increased mitochondrial membrane potential, intracellular ATP contents, mitochondrial mass and mitochondrial biogenesis, and reduced reactive oxygen species. Among the six flavan-3-ols, procyanidin B3 and procyanidin B1 exhibited the strongest effects. Our study suggests potential of proanthocyanidins as therapeutic target for diabetes.

## 1. Introduction

Diabetes mellitus (DM), characterized by hyperglycemia and deficits in insulin secretion or action, is one of the major chronic metabolic disorders [1]. It is estimated that about 700 million people worldwide will have diabetes by the mid of this century [2]. Most diabetic patients are type 2 diabetes mellitus (T2DM), with a major cause of deficient insulin signaling or insulin resistance [3]. It is well characterized that hyperglycemia exacerbates complications of diabetes [4]. In addition to hyperglycemia, hyperlipidemia is also regarded as one of the main risk factors of T2DM. Hyperlipidemia is a result of dysregulated hepatic lipid metabolism, which abnormally increases triglyceride (TG) synthesis and cholesterol production and/or decreases oxidative catabolism of fatty acid [5,6]. These abnormal metabolic processes lead to excessive fat accumulation in liver and induce hepatic steatosis. Therefore, improving lipid metabolism is a promising prevention and therapeutic strategy for T2DM. At present, the most prescribed medications for hyperlipidemia in diabetes include Fibrates and Stains. However, side effects of Fibrates and Stains [7,8,9] makes it significant to develop functional foods- and natural products-based treatments which could be safer and more tolerable for patients.

*Iris lactea* Pall. var. *Chinensis* (Fisch.) Koidz (*I. lactea*) is herbaceous perennials, belonging to the family of Iridaceae. *I. lactea* is widely distributed in China, India, Russia and South Korea and has been extensively used in traditional Chinese medicine [10]. *I. lactea* has very strong root system and is salt-tolerant and drought-resistant, which makes it suitable for greening and water/soil conservation in regions of dry climates [11]. In traditional Chinese medicine, the leaves of *I. lactea* are used to treat pharyngitis and joint pain, while the seeds, flowers, and roots are used for treatment of jaundice, diarrhea, leucorrhea and carbuncle swollen [12]. Irisquinone isolated from *I. lactea*, has been used in the treatment of different cancers, as both antineoplastic agent and radiosensitizer [13]. Moreover, recent studies reported that *I. lactea* possesses anti-radiation, anti-bacteria, anti-tumor and anti-oxidation activities [14,15,16,17,18]. Major constituents of *I. lactea* are suggested to be flavonoids, quinones, oligostilbenes, and fatty acids [19,20]. *I. lactea* contains proanthocyanidins, flavonoids polymer that are also known as condensed tannins [21]. Proanthocyanidins exist in forms of monomers and polymers. Monomers in proanthocyanidins are flavan-3-ols, such as catechin and epicatechin. Polymers in proanthocyanidins are formed by linkage of different numbers of monomers, mainly at C4, C6 and C8 positions [22]. Our lab has, for the first time, successfully purified six compounds (catechin, epicatechin, procyanidin B7, procyanidin B6, procyanidin B3 and procyanidin B1) from proanthocyanidins, with high speed counter-current chromatography. So far, proanthocyanidins have been shown to possess multiple biological functions, such as anti-oxidation, anti-inflammation, anti-tumor, anti-bacteria, sleep improvement, anti-hyperglycemia and lowering blood lipids [23,24,25]. However, there are few reports about the effects of proanthocyanidins in lipid metabolism dysregulation in T2DM, and the mechanism is far from being illustrated.

In the present study, we demonstrated the benefiting effects of proanthocyanidins on lipid metabolism in T2DM mice, and identified mechanism of inhibiting adipogenesis and protecting mitochondrial from damages in dexamethasone (Dex)-treated 3T3-L1 adipocytes.

## 2. Results

### 2.1. Spectrophotometric Analysis of the Proanthocyanidins

After testing of different concentrations of standard (+)-catechin solutions, the liner regression equation obtained in vanillin assay was Y = 1.7533X + 0.0415. The correlation coefficient was 0.9944. Precision of the spectrophotometric method was evaluated by intra-batch repetitive assay and the coefficient of variation was 5.1%, showing high precision for the spectrophotometric determination of total proanthocyanidins. Based on the calibration curve, we found the total proanthocyanidins content in the extracts was 542.30 ± 7.31 mg/g of the extracts.

### 2.2. Identification of Proanthocyanidins with UPLC-Triple-TOF/MS Analysis

UPLC-coupled online MS was used to characterize the presence of proanthocyanidins. Chromatogram at 280 nm is shown in Figure 1. All MS data were collected in 40 min run. Total ion chromatogram is shown in Figure 2 and details of MS data were summarized in Table 1. Proanthocyanidins are known as a big class of compounds containing different amounts of catechin or epicatechin units. In Figure 1, catechin and epicatechin are represented by peak 6 and peak 9, respectively. Area of proanthocyanidins peaks accounted for 31.44% of that of total peaks. MS/MS data confirmed catechin (*m*/*z* 289) and epicatechin (*m*/*z* 289) as the major proanthocyanidins components. Besides proanthocyanidins, stilbene compounds were also detected, with elution time of 20–24 min. Area of stilbene peaks accounted for 18.01% of total area. In this study, we focused on proanthocyanidins, the major constituents of which (catechin, epicatechin, procyanidin B1, procyanidin B3, procyanidin B6 and procyanidin B7) have been separated by previous studies [26,27].

### 2.3. Effects of Proanthocyanidins on Body Weight and Metabolic Markers in Serum

To investigate the effects of proanthocyanidins on T2DM, we treated mice with proanthocyanidins after induction of T2DM with high-fat-diet (HFD) feeding and streptozocin (STZ) injection. As shown in Figure 3a, the body weights of all T2DM groups declined from week 0 (the week that proanthocyanidins administration started) to week 1, regardless of the treatment. In contrast, body weights of normal control mice (NG, fed with normal chow and received injection of citrate buffer instead of STZ) maintained at a normal level. From week 2 to week 4, body weights of MG mice (model group, T2DM mice that received vehicle administration) was significantly increased, compared to NG group. Treatment of fenofibrate (Fen) or low and high dosages of proanthocyanidins (referred to as PC-L and PC-H, respectively) reduced body weight of T2DM mice, compared to MG group. As shown in Figure 3b, glucose levels of MG group were significantly higher than that of the NG group. After 6 weeks of treatment, the glucose levels gradually declined in Fen, PC-H, and PC-L groups, compared to MG group. Interestingly, PC-L displayed stronger effect in alleviating hyperglycemia than PC-H. Moreover, as shown in Figure 3c, levels of insulin in serum of MG group were higher than the NG group, while treatment with Fen, PC-L, or PC-H reduced the level of insulin in serum. These results suggest that proanthocyanidins can ameliorate hyperglycemia and hyperinsulinemia in diabetes, which effect is comparable to that of fenofibrate.

Next, we analyzed lipid levels in serum to determine the effects of PC-H and PC-L on dyslipidemia. As shown in Figure 3d–g, mice of MG group exhibited elevated triglycerides (TG), total cholesterol (TC) and low-density lipoprotein-cholesterol (LDL-C), compared to that of NG group. Treatment with PC-H or PC-L reduced the levels of TG, TC, and LDL-C, compared to MG group. It should be noted that proanthocyanidins can restore the LDL-C to a level that is comparable to the NG group. Biochemical markers of liver damage were also measured. As shown in Figure 3h–i, compared to NG group, levels of ALT and AST in MG group were slightly increased. Fen treatment increased the levels of ALT and AST, suggesting liver damaging effect of Fen. In contrast, proanthocyanidins lowered levels of AST although ALT levels remained high, compared to MG group. In summary, attenuation of HFD/STZ-induced changes in TG, TC, LDL-C and AST after proanthocyanidins treatment suggests its beneficial effect in treatment of hyperlipemia and liver damage in diabetes.

### 2.4. Effects of Proanthocyanidins on Expression of Lipid Metabolism-Associated Proteins

To further identify the mechanism of proanthocyanidins attenuating hyperlipidemia in diabetic mice, the lipid metabolism-related proteins were analyzed with western blot. Hepatic lipogenic and lipolytic proteins including FAS, ACC, ATGL, HSL, and CPT1A were detected. Phosphorylation of AMPK was also detected, considering its role as a sensor of energy metabolism [28]. ACC is a key regulator in lipid synthesis, which converts acetyl-CoA to malonyl-CoA. FAS modulate lipid homeostasis, increased expression of which leads to hepatic TG accumulation [29]. The activation of ACC and FAS inhibits CPT1A, which enhances fatty acid β-oxidation in mitochondria. As shown in Figure 4a–c, levels of FAS and ACC in diabetic mice were significantly increased compared to that of NG group. Compared with MG group, FAS and ACC expression in PC-H and PC-L groups were dramatically reduced. It is well known that ATGL is a triglyceride lipase that primarily catalyzes the first step of triglyceride hydrolysis. HSL is a rate-limiting lipolytic enzyme that exhibits higher substrate affinity for diacylglycerol. CPT1A enhances the transportation of fatty acid into mitochondrial and is involved in fatty acid β-oxidation. Western blot results showed reduced expression of ATGL and HSL in MG group, compared to NG group. Expression of CPT1A in MG group showed a trend of reduction, compared to NG group, although no statistical significance was obtained. These data indicated that fat hydrolysis and mitochondrial function in diabetic mice is severely impaired. After proanthocyanidins treatment, expression of lipolysis-related proteins were up-regulated. Taken together, these results indicate that proanthocyanidins might ameliorate HFD/STZ-induced hypertriglyceridemia via inhibiting lipogenesis and promoting lipid hydrolysis and fatty acid β-oxidation. Furthermore, as shown in Figure 4c, p-AMPK/AMPK ratio in liver tissue decreased in MG mice, compared to NG mice. 6-week treatment with proanthocyanidins up-regulated phosphorylation of AMPK in diabetic mice. PC-H-induced up-regulation is of statistical significance. These results suggest that proanthocyanidins may ameliorate deficits of hepatic lipid metabolism in HFD/STZ-induced diabetic mice through activating the AMPK/ACC/CPT1A signaling pathway.

### 2.5. Effects of Flavan-3-Ols on Intracellular Lipid Accumulation in 3T3-L1 Adipocytes

We investigated the anti-adipogenic effects of the six flavan-3-ols (Figure 5a) in cultured 3T3-L1 adipocytes. First, we examined the effects of six flavan-3-ols on cell viability in 3T3-L1 preadipocytes via MTT assay. As shown in Figure 5b, cell viability of 3T3-L1 preadipocytes was not altered after incubation with flavan-3-ols for 48 h at concentrations of 0.1, 1 and 10 µM. Decrease in cell viability was observed when concentration increased to 50 µM. Lipid accumulation was determined by oil-red-O staining at the end of differentiation on Day 8. Among all treatment, the PB3 and PB1 groups showed lowest accumulation of lipid droplets (Figure 5c). Cellular TG content were determined by TG kit and reduction in TG content were induced after flavan-3-ols treatment, with PB3 and PB1 showing most potent effect (Figure 5d).

### 2.6. Effects of Flavan-3-Ols on Adipogenesis

Previous studies suggested that PPARγ and C/EBPα are key transcriptional factors, which are necessary and sufficient for adipocyte differentiation and adipogenesis [30]. FABP4 has been widely used as a marker of differentiated adipocytes. To determine the effects of flavan-3-ols on the expression of adipogenesis, western blot and RT-PCR analysis were performed to analyze the expression of PPARγ, C/EBPα and FABP4. As shown in Figure 6a, protein levels of PPARγ, C/EBPα and FABP4 in the differentiated 3T3-L1 cells (diff) were strongly increased, compared to the undifferentiated 3T3-L1 cells (undiff). Similar changes were observed in the mRNA levels of PPARγ, C/EBPα and FABP4 (Figure 6b–d). Treatment with flavan-3-ols inhibited the expression of PPARγ, C/EBPα and FABP4 at both protein levels and mRNA levels, when compared to the un-treated differentiated adipocytes. Strongest effect was observed in PB3- or PB1-treated group (Figure 6a–d). These results suggest that flavan-3-ols in proanthocyanidins possess anti-adipogenic effect in 3T3-L1 adipocytes, possibly via suppressing PPARγ, C/EBPα and FABP4 expression.

### 2.7. Effects of Flavan-3-Ols on Glucose Uptake and Phosphorylation of AKT in Insulin-Resistant 3T3-L1 Adipocytes

Effects of flavan-3-ols on glucose uptake in 3T3-L1 adipocytes of insulin resistance are shown in Figure 7. 48 h pretreatment with 1 µM Dex in fully differentiated 3T3-L1 cells decreased insulin (100 nM, 15 min)-induced glucose uptake, compared to non Dex-treated adipocytes (Figure 7a). Treatment with Rosi, PB1 or PB1 increased the glucose uptake in Dex-pretreated cell (Figure 7a). Additionally, analysis of AKT phosphorylation in insulin-resistant 3T3-L1 adipocytes showed a suppressing effect from Dex. Treatment with Rosi or flavan-3-ols attenuated Dex-induced suppression of glucose uptake and AKT phosphorylation in differentiated 3T3-L1 cells. Among the flavan-3-ols, PB3 and PB1 increased both glucose uptake and AKT phosphorylation in Dex-pretreated 3T3-L1 adipocytes (Figure 7b). These results indicated that insulin resistance in differentiated 3T3-L1 adipocytes could be induced by Dex treatment, while flavan-3-ols attenuated Dex-induced deficits in glucose uptake, perhaps via increasing AKT phosphorylation.

### 2.8. Effects of Flavan-3-Ols on Mitochondrial Function in Insulin-Resistant 3T3-L1 Adipocytes

It has been reported that mitochondrial dysfunction plays important roles in insulin resistance and T2DM [31]. To further demonstrate their treatment effects on T2DM, we evaluated the protective effect of flavan-3-ols on mitochondrial functions in Dex-treated 3T3-L1 adipocytes. Differentiated 3T3-L1 adipocytes were incubated with Dex, with or without presence of flavan-3-ols, for 48 h. After that, the mitochondrial superoxide production, mitochondrial membrane potential (MMP), intracellular ATP contents, and mitochondrial mass were analyzed. As shown in Figure 8a–d, Dex treatment decreased MMP level, ATP contents, mitochondrial mass while increased mitochondrial superoxide production in 3T3-L1 adipocytes. Mitotempo is a mitochondrial-targeted antioxidant that can scavenge superoxide. Cccp (carbonyl cyanide 3-chlorophenylhydrazone) is a potent mitochondrial oxidative phosphorylation uncoupler, leading to loss of MMP. As positive controls, mitotempo and cccp significantly reduced mitochondrial superoxide and MMP in insulin-resistant 3T3-L1 cells, in agreement with previous observations [32,33]. When Dex-incubated adipocytes were treated with flavan-3-ols, the MMP levels, ATP contents, and mitochondrial mass increased, while mitochondrial superoxide accumulation decreased. Strongest effect was observed after treatment with PB3 and PB1.

### 2.9. Effects of Flavan-3-Ols on Expression of Mitochondrial Biogenesis-Related Proteins in Insulin-Resistant 3T3-L1 Adipocytes

PGC-1α is critical in regulating mitochondrial biogenesis. NRF1 is a nuclear gene-encoded transcription factor that target to mitochondria and induces expression of Tfam [33]. Tfam can promote replication of mitochondrial DNA and transcription of mitochondrial genes. We investigated the effects of flavan-3-ols on expression of PGC-1α, NRF1, and Tfam in insulin-resistant 3T3-L1 adipocytes. As shown in Figure 9, Dex treatment caused significant reduction in the mRNA levels of PGC-1α, NRF1, and Tfam, while treatment with Rosi or compounds PB3 and PB1 increased their expression. On the other hand, treatment with flavan-3-ols in insulin-resistant 3T3-L1 adipocytes also increased the protein levels of PGC-1α and SirT1, which deacetylates and activates PGC-1α [34] (Figure 9d). In particular, compound PB1 displayed the strongest effect in elevating protein levels of SirT1 and PGC-1α (Figure 9d).

### 2.10. Effects of Flavan-3-Ols on Mitochondrial Dynamics and Mtdna Damage in Insulin-Resistant 3T3-L1 Adipocytes

Mitochondria undergo continuous fusion and fission, balance of which is critical for maintaining their morphological and functional normality [35]. To determine the effects of flavan-3-ols on mitochondrial dynamics, western blot analysis was performed to determine the levels of mitochondrial fission proteins (Drp1) and mitochondrial fusion proteins (Mfn2 and Mfn1). Results showed that Dex treatment selectively increased the protein levels of Mfn2 and Drp1, while Mfn1 was not altered (Figure 10a). Treatment with flavan-3-ols showed different effects on the expression of Drp1, Mfn2, and Mfn1. Compound PB1 significantly decreased the levels of Drp1 and increased that of Mfn2, while compound PB3 significantly increased the expression of both Mfn2 and Mfn1. Damage of mitochondrial DNA (mtDNA) was detected by long PCR. As shown in Figure 10b, Dex caused significant mtDNA damage in 3T3-L1 adipocytes, while treatment with compound EC, PB6, or Rosi attenuated the damage.

## 3. Discussion

Proanthocyanidins have been suggested to benefit various metabolic disorders. Grape seed-derived proanthocyanidin alleviated oxidative stress and ER stress in diabetic conditions caused by HFD/STZ [36]. A-type procyanidins from litchi pericarp attenuated hyperglycemia in the same model [37]. In HFD-fed rats, proanthocyanidins extracted from grape seed corrected dyslipidemia via repressing VDLD assembling in liver [38]. However, mechanism underlying the beneficial effects of proanthocyanidins remains unclear, which limited their pharmacological utility.

HFD feeding combined with intraperitoneal STZ injection is a well-accepted method to establish T2DM model [39]. The mechanism might be HFD-induced glucose intolerance and insulin resistance and STZ caused severe damage of pancreatic β cells [40]. Besides the deficits in glucose metabolism, dysregulated lipid metabolism, such as hyperlipidemia and hepatic steatosis are also closely associated with T2DM [41]. Dysregulated lipid metabolism in liver causes triglycerides accumulation and induces insulin resistance, contributing to development of diabetes. In this study, proanthocyanidins improved body weight and serum lipid profiles (Figure 3). Regarding the mechanism, both PC-L and PC-H decreased the lipid accumulation via modulation expression of lipogenesis- and lipolysis-related proteins in liver (Figure 4), in an AMPK-mediated manner. AMPK has been recognized as a promising target for alleviating metabolic syndromes, which can directly phosphorylating metabolic enzymes, including hepatic lipogenesis- and lipolysis-related proteins [42]. Taken together, our results demonstrated that proanthocyanidins ameliorated deficits in hepatic lipid metabolism in HFD/STZ-induced diabetic mice, perhaps through the AMPK/ACC/CPT1A signaling pathway.

Adipose tissue is an important insulin-sensitive endocrine organ and plays critical role in regulating metabolism of the whole body [43]. In the present study, we investigated the effects of proanthocyanidins-derived flavan-3-ols on metabolism of cultured adipocytes. First, we studied the effects of flavan-3-ols on adipogenesis in 3T3-L1 cells. Measurement of TG content and oil-red-O staining showed that all the six flavan-3-ols inhibited accumulation of lipid droplet, with PB3 and PB1 possessing strongest effect (Figure 5). To identify the molecular mechanism, we detected expression of PPARγ, C/EBPα and FABP4, factors that promote adipogenesis. Western blot and RT-PCR results showed reduced protein and mRNA levels of PPARγ, C/EBPα and FABP4 after treatment of flavan-3-ols, with a possible stronger effect in PB3 and PB1 groups (Figure 6). Taken together, our results demonstrated that flavan-3-ols inhibited adipogenesis of 3T3-L1 adipocytes via down-regulating expression of adipogenesis-related genes, and suggest the potential of PB3 and PB1 in treating obesity.

Insulin resistance is a major causative factor in T2DM. Defects in insulin signaling and disordered glucose and lipid metabolism account for the progression of insulin resistance [44]. Liver, adipose tissue and skeletal muscles are the major peripheral organs in regulating lipid and glucose metabolism and play important roles in controlling insulin sensitivity [45]. In the present study, we established a model of insulin resistance by treating 3T3-L1 adipocytes with Dex. The development of insulin resistance was verified by decreased 2-NBDG uptake and AKT phosphorylation (Figure 7). Interestingly, flavan-3-ols incubation increased glucose uptake and AKT phosphorylation in insulin-resistant 3T3-L1 adipocytes (Figure 7). Among the six compounds, PB3 and PB7 exhibited the strongest effect.

Mitochondrial dysfunction greatly contributes to insulin resistance and T2DM [46]. We studied the effects of flavan-3-ols on Dex-induced mitochondrial dysfunction in 3T3-L1 adipocytes. Consistent with previous studies [47], Dex caused accumulation of mitochondrial superoxide, decrease of MMP, reduction of ATP synthesis and mitochondrial mass, in differentiated 3T3-L1 adipocytes (Figure 8). Treatment with flavan-3-ols or Rosi restored these parameters in Dex-treated 3T3-L1 adipocytes (Figure 8).

Mitochondrial biogenesis is critical in maintaining the normal physiological function of mitochondria. Mitochondrial biogenesis is strictly regulated by different modulators, including SirT1 and PGC-1α. PGC-1α is a coactivator of transcription factors that regulates expression of mitochondrial genes including NRF1 and Tfam [38], while SirT1 deacetylates and activates PGC-1α [34]. Our results showed that Dex treatment decreased expression of mitochondrial biogenesis-associated genes (Figure 9) and induced mtDNA damage (Figure 10) in 3T3-L1 adipocytes. Interestingly, treatment with flavan-3-ols or Rosi significantly increased the expression of mitochondrial biogenesis-associated genes and attenuated mtDNA damage (Figure 9 and Figure 10). It has been reported that mitochondria undergo consecutive fission and fusion to maintain its normal morphology and function [48]. Dynamics of mitochondria are under fine control of modulators including the fusion proteins (such as Mfn1 and Mfn2) and fission proteins (such as Drp1 and Fis1) [49]. Fudion/fission imbalance causes mitochondria fragmentation, probably due to accumulated reactive oxygen species (ROS) and mtDNA damage [50]. The western blot results suggest that Dex treatment caused increase Drp1 expression (Figure 10), while treatment with PB1 reduced the expression of Drp1 and increased that of Mfn2 (Figure 10).

In earlier studies, we found that natural nutritional products, such as isoorientin isolated fenugreek seeds, stimulated mitochondrial biogenesis and reduced oxidative stress to improve mitochondrial function and ameliorated insulin resistance [51]. Researchers define mitochondrial nutrients as those that protect the mitochondria from oxidative damage and improve mitochondrial function [52]. These nutrients might protect mitochondrial through (a) inhibiting or preventing oxidant production and/or scavenging free radicals and ROS to eliminate oxidative stress in mitochondria, such as α-tocophenol and lipoic acid [53]; (b) acting as phase 2 enzymes inducers to repair mitochondrial damage and enhance antioxidant defenses, such as sulforaphane [54]; (c) serving as cofactors/substrates to protect mitochondrial enzymes and/or stimulate mitochondrial enzyme activity, such as B vitamins [53]; (d) enhancing mitochondrial metabolism to increase mitochondrial biogenesis, such as alpha lipoic acid and acetyl-L-carnitine [55,56]. C and EC are representatives of flavan-3-ols, a group of polyphenolic compounds [57]. Furthermore, C, EC and their most common dimers, including PB7, PB6, PB3, PB1, are the main biologically active compounds present in proanthocyanidins. Early studies demonstrated that monomeric cocoa catechin promotes β-cell stability through up-regulating the expression of key genes that encode antioxidant and mitochondrial respiratory complex proteins [58]. In addition, other studies shown that epicatechin and procyanidin B2 directly influence mitochondrial functions by reducing release of cytochrome c in isolated rat heart [59]. In line with these reports, our results demonstrated that C, EC, PB7, PB6, PB3, and PB1 ameliorated Dex-induced mitochondrial deficits in 3T3-L1 adipocytes, via improving mitochondrial biogenesis, dynamics, transmembrane potential, and antioxidant defenses.

## 4. Materials and Methods

### 4.1. Chemicals and Reagents

The dried seeds of *I. lactea* were collected from Malian Lake of Alxa League of Inner Mongolia, China. The plant was identified by Dr Yuhu Wu (Northwest Institute of Plateau Biology, Chinese Academy of Sciences, China) and authenticated using the voucher specimen in the Qinhai–Tibeta Plateau Museum of Biology (Xining, China) (reference No.158719) and the Chinese Virtual Herbarium (http://qtpmb.cvh.org.cn). Commercial silica gel (020160910, Qing Dao Hai Yang Chemical Group Co., 200–300 mesh) was used for column chromatography. Vanillin and (+)-catechin were purchased from Shanghai yuanye Bio-Technology Co. Ltd. (Shanghai, China). Biochemical kits of serum glucose, triglycerides (TG), total cholesterol (TC), high-density lipoprotein-cholesterol (HDL-C), low-density lipoprotein-cholesterol (LDL-C), alanine aminotransferase (ALT) and aspartate aminotransferase (AST) were purchased from Nanjing Jiancheng Bioengineering Institute (Nanjing, China). Enzyme-linked immunosorbent assay (ELISA) kit for detection of Mouse Insulin and ATP assay kit were purchased from Beyotime Biotechnology (Shanghai, China). Dulbecco’s modified Eagle’s medium (DMEM) and fetal bovine serum (FBS) were obtained from Gibco (Carlsbad, CA, USA). Streptozocin (STZ), Isobutyl-3-methyl-xanth-ine (IBMX), Dexamethasone (Dex), MTT, oil-red-O, fluorescent probe of 2-NBDG and JC-1 were purchased from Sigma-Aldrich Chemical Co. Ltd. (St Louis, MO, USA). Mitosox and mitotracker green were purchased from Thermo Fisher Scientific (Rockford, IL, UAS). Biosynthetic Human Insulin Injection was purchased from Novo Nordisk A/S (Novo Alle, Bagsvaerd, DK). Pierce BCA Protein Assay Kit was purchased from Thermo Fisher Scientific (Rockford, IL, USA). The DNA extraction kit was purchased from TIANGEN Biotech Co. Ltd. (Beijing, China). The total RNA extraction kit was purchased from Thermo Fisher Scientific (Rockford, IL, USA). Antibodies of β-actin, FAS, ACC, ATGL, HSL, CPT1A, p-AMPK, AMPK, PPARγ, C/EBPα, FABP4, AKT, p-AKT, Mfn1, Mfn2, Drp1, SirT1 and PGC-1α were purchased from Cell Signaling Technology (Danvers, MA, USA). All chemicals being used are of reagent grade.

### 4.2. Preparation of Enriched Proanthocyanidins

Proanthocyanidins were extracted from dried seed coats of *I. lactea* by supercritical CO_2_ in a HA221-40 (50)-(10 + X) supercritical extractor (Nantong Yichuang Experimental Apparatus Co. Ltd., Nantong, China). The supercritical CO_2_ parameters were extraction pressure, 35 MPa; extraction temperature, 50 °C; extraction time, 2.5 h; mass flow rate, 5 L/h; characteristic particle size, 40–60 mesh. The residue was extracted with 75% ethanol at 60 °C for three times, with 3 h each time. The extracts were combined and underwent concentration under reduced pressure. Concentrated extracts were then dispersed in water and extracted with ethyl acetate and n-butanol successively. Extracts in ethyl acetate and n-butanol extracts were combined and concentrated under reduced pressure. Finally, three portions of extracts in ethyl acetate phase, n-butanol phase, and aqueous phase were obtained. The ethyl acetate fraction was loaded onto silica gel column and eluted with gradient mobile phase composed of light petroleum and ether-ethyl acetate (from 5:4 to 5:5). The eluted fractions were collected as enriched proanthocyanidins. Isolation and purification of the six flavan-3-ols from proanthocyanidins were performed as previously described [26,27]. The enriched proanthocyanidins and its six flavan-3-ols compounds, catechin (C), epicatechin (EC), procyanidin B7 (PB7), procyanidin B6 (PB6), procyanidin B3 (PB3), procyanidin B1 (PB1) were stored at −20 °C.

### 4.3. Spectrophotometric Analysis

Total contents of proanthocyanidins were determined with vanillin assay [60,61,62]. Standard curve and calibration curve were obtained by measuring absorption of different concentration of standards solution. Concentration of (+)-catechin stock solution was 1 mg/mL, dissolved in methanol. Stock solutions were diluted into series of concentrations: 0, 0.02, 0.04, 0.06, 0.08, 0.1 mg/mL. Different concentrations of (+)-catechin were reacted with vanillin following procedure below. 0.0100 g proanthocyanidins were dissolved with 25 mL of methanol; 1mL of proanthocyanidins-methanol sample was added to 6 mL vanillin (4% in methanol); adding 3 mL 37% HCL under shaking to acidify the sample. After incubation at 30 °C for 30 min under protection from direct light exposure, the mixture was measured for absorption at 500 nm. The concentration of proanthocyanidins in tested samples was determined according to the standard curve.

### 4.4. HPLC Analysis

For HPLC analysis, 20 mg of proanthocyanidins were dissolved in 2 mL of 50% acetonitrile and processed by ultrasonic baths (30 min), and 0.22 µm filtration. Samples were analyzed with Agilent ZORBAX-SB HPLC system equipped with ultraviolet-visible (UV-vis) detector and a C18 column (100 mm × 4.6 mm, Agilent, USA). 5 µL aliquot of proanthocyanidins/acetonitrile solution was loaded. The mobile phase was 0.1% methanoic acid in water (A) and 0.1% methanoic acid in acetonitrile (B). Gradient elution was performed as follows: 0–2 min, 95% A; 2–25 min, 95–50% A; 25–35 min, 50–5% A; 35–37 min, 5% A; 37–40 min, 5–95% A. The flow rate was 0.8 mL/min, and temperature was 30 °C. Detection of components was performed at 280 nm.

### 4.5. UPLC-Triple-TOF/MS

An ACQUITY UPLC detecting system (Waters Co., Milford, MA, USA) coupled to a triple time-of-flight (TOF) 5600+ mass spectrometer were employed for UPLC-Triple-TOF/MS analysis. An electrospray ionization source (AB SCIEX Co., Foster, CA, USA) and a ZORBAX-SB C18 analytical column (100 mm × 4.6 mm, Agilent Technologies Inc., Santa Clara, CA, USA) were used. Gradient elution conditions were the same as described above. MS conditions were listed as follows: negative ion scanning mode, MS scanning range of 100–1500 m/z; gas1 (GS1): 50 psi; gas2 (GS2): 50 psi; curtain gas (CUR): 35 psi; temperature of the ion source (TEM): 550 °C; voltage of the ion source(IS): −4500 V (negative); first order scanning: declustering potential (DP): 100 V; focusing voltage (CE): 10 V; second order scanning: TOF MS~Product Ion~IDA mode for data collection. CID energy: −20, −40 and −60 V. Before sample injection, CDS pump were used for mass axis correction to lower the error below 2 ppm.

### 4.6. Animals and Treatments

Six-week-old ICR male mice (20 ± 2 g) were obtained from Jinan Pengyue experimental animal breeding Co. Ltd. (Jinan, China). Animals were housed in environment of temperature 25 ± 2 °C, humidity 55 ± 5% and 12 h light/dark cycle, with free access to tap-water and standard food. All experimental procedures involving the usage of animals were approved by the Institutional Animal Care and Use Committee of the Chinese Academy of Science, Northwest Institute of Plateau Biology. Prior to experiment, ICR mice could acclimate to the environment for one week. The T2DM mouse model was induced as previously reported [63]. Design of animal experiment and treatment is summarized in Figure 11. After acclimation, some mice were fed with normal chow (3.2 kcal/g body weight, 4.65% calories derived from fat) and received intraperitoneal injection of citrate buffer (referred to as normal group, NG). The other mice were fed with HFD, 5 kcal/g body weight, 60% calories derived from fat) and received intraperitoneal injection of STZ (dissolved in citrate buffer, pH 4.4–4.5, 50 mg/kg, once per day for 3 consecutive days) to induce diabetes at week 9. At week 10, serum glucose levels were measured and those with value higher than 11.0 mM were used as diabetes mice for following studies. Diabetes mice were divided into four groups, the model group (vehicle treated, also referred to as group 2), Fen group (treated with 250 mg/kg fenofibrate, group 3), PC-H group (treated with 200 mg/kg proanthocyanidins, group 4) and PC-L (treated with 50 mg/kg proanthocyanidins, group 5). During treatment, mice were fed with HFD. The rodent food was purchased from Trophic Animal Feed High-Tech Co. Ltd. (Nantong, China). Drug administration was conducted via oral gavage between 8:30 and 9:30 o’clock in the morning from week 10 to 16, and the dosages were daily adjusted according to the body weight. At week 16, blood samples were obtained via retro-orbital sinus, and serum was collected after centrifugation (5000 g) for 15 min at room temperature. Mice were then sacrificed, and liver tissue was carefully dissected, washed with ice-cold saline, frozen with liquid nitrogen and kept at −80 °C.

### 4.7. Biochemical Analysis of Blood Samples

After the last gavage administration, all mice were fasted for 12 h. Then the blood samples were collected, and serum levels of glucose, TG, TC, HDL-C, LDL-C, ALT, and AST were measured with commercial kits, following manufacturers’ instructions. The levels of insulin in serum were determined by ELISA kit.

### 4.8. Western Blot Analysis

Total proteins were extracted from liver tissues or 3T3-L1 adipocytes with RIPA buffer containing protease inhibitors. After centrifugation at 12,000 g at 4 °C for 15 min, the supernatants were collected and stored at −80 °C. Protein concentration was measured with BCA Protein Assay Kit. 30 µg/µL protein samples were separated on 6% and 10% SDS polyacrylamide gel and transferred to nitrocellulose membrane. The membrane was incubated with primary antibodies overnight at 4 °C. After washing with 1% TBST, the membrane was incubated with peroxidase-conjugated secondary antibodies for 1 h at room temperature. Western blot bands were visualized with 5200 Multi Luminescent imaging systems and an ECL detection kit. Results were analyzed with Image J, with β-actin as inner control.

### 4.9. Cell Culture and Treatment

Mouse 3T3-L1 preadipocytes were purchased from the cell bank of the Institute of Biochemistry and Cell Biology of Shanghai (Shanghai, China). The preadipocytes were grown in DMEM containing 10% FBS and antibiotics at 37 °C in a humidified incubator with 5% CO_2_. For induction of differentiation, 3T3-L1 preadipocytes were incubated in DMEM medium (containing 10% FBS, 10 µg/mL insulin, 1 µM Dex and 0.5 mM IBMX) for 2 days after confluence (defined as day 0). Cells were then incubated with 10% FBS in DMEM and 10 µg /mL insulin for 2 days. Starting at Day 4, the medium was replaced by fresh DMEM every 2 days until Day 8. To investigate the effects of compounds on 3T3-L1 preadipocytes differentiation, cells were cultured in differentiation medium and treated with flavan-3-ols on days 0–4. Fully differentiated 3T3-L1 adipocytes were treated with 1 µM dexamethasone (Dex) for 48 h. Successful induction of insulin resistance was confirmed by decrease in insulin-induced uptake of 2-(N-(7-Nitrobenz-2-oxa-1,3-diazol-4-yl) Amino)-2-Deoxyglucose (2-NBDG). To examine the effects of flavan-3-ols on insulin resistance and mitochondrial function, related parameters were measured after 48 h co-incubation of Dex and flavan-3-ols in 3T3-L1 adipocytes.

### 4.10. MTT Assay

3T3-L1 preadipocyte cells were seeded in 96-well plates at a density of 1 × 10^4^ cells per well in DMEM containing 10% FBS. Cells were then exposed to various concentrations (0–100 µM) of flavan-3-ols for 48 h. After that, MTT solution (5 mg/mL) was added and incubated at 37 °C for 4 h in the dark. The medium was removed and 100 µL of DMSO was added. Light absorbance at 490 nm was determined by a microplate reader (Molecular Devices, Sacramento, CA, USA).

### 4.11. Measurement of Cellular Lipid Accumulation and Lipolysis

Cellular lipid droplets were detected by oil-red-O staining. Cells were washed twice with phosphate buffered saline (PBS) and fixed in 4% paraformaldehyde for 30 min. Then, each well was washed with PBS three times and stained with oil-red-O for 1 h in the dark. The stained lipid droplets in 3T3-L1 cells were photographed by light microscopy (Olympus, Tokyo, Japan). The content of intracellular triacylglycerol was determined with a commercial assay kit, following the manufacturer’s instructions. Protein concentration was measured by BCA Protein Assay Kit.

### 4.12. Glucose Uptake Assay

Glucose uptake assay was performed in accordance with previous study [51]. The fully differentiated 3T3-L1 adipocytes were pre-treated with 1 µM Dex for 48 h on 12-well plates. After stimulation with insulin (100 nM, 15 min), culture medium was removed and glucose-free culture medium containing 100 µM of 2-NBDG was added. Cells were incubated at 37 °C for 30 min. After that, incubation medium was removed, and cells were washed twice with pre-cooled PBS to stop uptake of 2-NBDG. Cells were scanned by NovoCyte 2040R flow cytometer (ACEA, Hangzhou, China) (Ex = 485 nm, Em = 535 nm).

### 4.13. Measurement of Mitochondrial Functions

3T3-L1 cells were cultured in 12-well culture plate at density of 5 × 10^4^ cells per well. Measurements of intracellular mitochondrial superoxide levels and MMP were performed with fluorescence dye Mitosox and mitochondrial-2 specific lipophilic cationic staining kit (JC-1), respectively. The fluorescence intensity was quantified by NovoCyte 2040R flow cytometer. The intracellular contents of ATP were determined by a commercial kit of bioluminescent method. Quantification of mitochondrial mass in 3T3-L1 adipocytes was performed after staining with mitotracker green [64]. Cells were first incubated with 100 nM mitotracker green for 30 min in the dark. After that, cells were washed three times with ice-cold PBS and visualized by confocal laser scanning microscopy (TE2000, Nikon, Japan).

### 4.14. RNA Isolation and Reverse Transcription PCR

Total RNA was isolated from 3T3-L1 adipocytes using Trizol reagent (Invitrogen, Carlsbad, CA, USA), in accordance with the manufacturers’ instruction. The first-strand cDNA was generated using oligo (dT) primer and the mRNA levels were determined by quantitative real-time PCR analysis [65]. The reaction was conducted in 7500 fast Real-Time PCR System (Applied Biosystems, Foster, CA, USA). Data were analyzed with comparative critical threshold (Ct) method, with β-actin as the inner control. Sequences of primers used are listed in Table 2.

### 4.15. DNA Isolation and Long PCR

Total DNA was isolated from 3T3-L1 adipocytes using DNA extraction kit. Based on the idea that DNA damage blocks DNA polymerase progression [66]. PCR was employed in detection of DNA damage. The long fragment target DNA was amplified with Long Amp Taq Master Mix polymerase, while the short fragment target DNA was amplified with TaKaRa Taq DNA polymerase (primer sets listed in Table 2). The PCR reactions were performed in Veriti 96 well thermal cycler long PCR system (Applied Biosystems, Foster, CA, USA) as previously described [51]. After amplification, the PCR products were resolved in 1.0% and 0.5% agarose gel and visualized by staining with ethidium bromide. The long/short mtDNA ratio was calculated based on intensity of bands.

### 4.16. Statistics

All values are expressed as Mean ± SD. The GraphPad Prism 7.0 (GraphPad Software, Inc., La Jolla, CA, USA) was used for statistical analysis. For in vivo studies, statistical significance was determined by using a two-tailed unpaired Student’s test or One-way ANOVA. For in vitro studies, statistical significance was determined by One-way ANOVA. Difference with P-values less than 0.05 were considered to be significant.

## 5. Conclusions

In conclusion, the present study demonstrated that proanthocyanidins from *I. lactea* ameliorated lipid metabolism and attenuated hepatic steatosis in mice with HFD/STZ-induced T2DM, for which activation of AMPK/ACC/CPT1A signaling might be an underlying mechanism. Additionally, we revealed that flavan-3-ols of proanthocyanidins extracted from *I. lactea* inhibited adipogenesis in 3T3-L1 adipocytes, and ameliorated Dex-induced insulin resistance and mitochondrial dysfunctions. Among the flavan-3-ols compounds, PB3 and PB1 exhibited the highest effects. Our study suggests proanthocyanidins may benefit treatment of type 2 diabetes and hypertriglyceridemia.

## Figures and Tables

**Figure 1 ijms-21-02029-f001:**
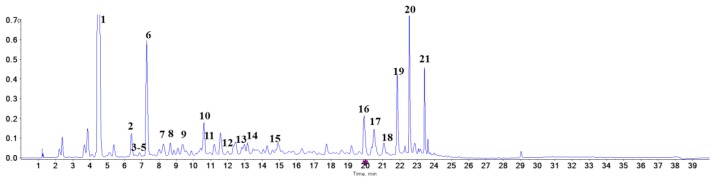
HPLC chromatogram of proanthocyanidins from *Iris lactea* Pall. var. *Chinensis* (Fisch.) Koidz at 280 nm.

**Figure 2 ijms-21-02029-f002:**
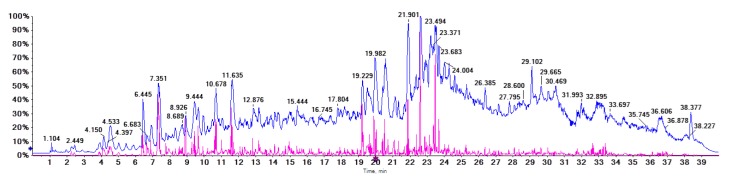
Total ion chromatogram (TIC) of *Iris lactea* Pall. var. *Chinensis* (Fisch.) Koidz in negative ion mode.

**Figure 3 ijms-21-02029-f003:**
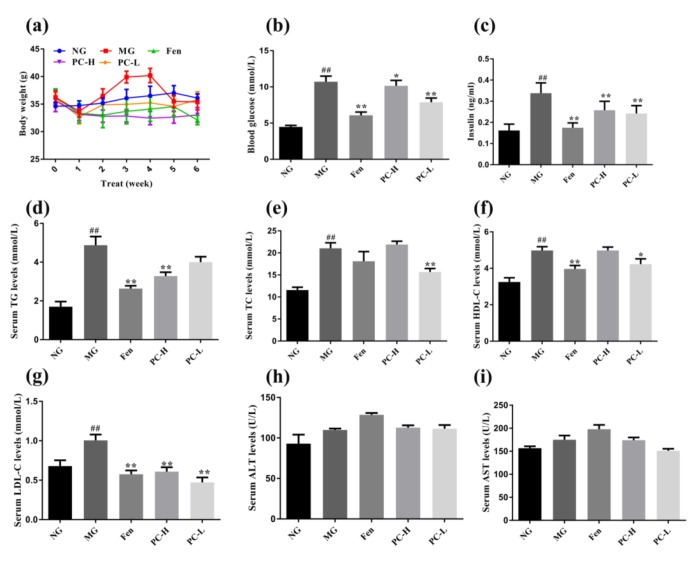
Effects of proanthocyanidins on serum biomarkers in HFD/STZ-induced type 2 diabetic mice. (**a**) Body weight. (**b**) Glucose level. (**c**) Insulin content. (**d**) TG. (**e**) TC. (**f**) HDL-C. (**g**) LDL-C. (**h**) ALT. (**i**) AST. NG, normal diet group; MG, HFD/STZ-induced T2DM; Fen, HFD/STZ-induced T2DM + 250 mg/kg fenofibrate; PC-H, HFD/STZ-induced T2DM + 200 mg/kg proanthocyanidins; PC-L, HFD/STZ-induced T2DM + 50 mg/kg proanthocyani-dins. Values are expressed as Mean ± SD, with 9 mice in each group. # *p* < 0.05 and ## *p* < 0.01 compared with the NG group; * *p* < 0.05 and ** *p* < 0.01 compared with the MG group.

**Figure 4 ijms-21-02029-f004:**
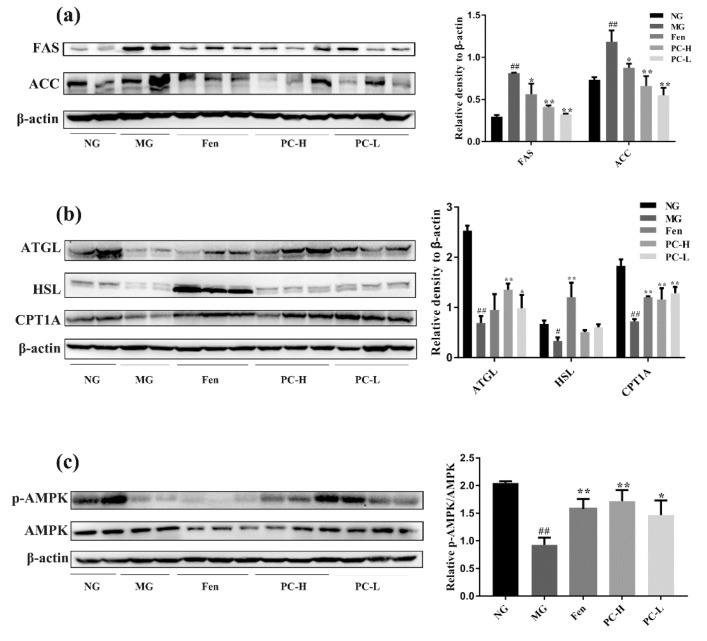
Effects of proanthocyanidins on expression of lipid metabolism-associated proteins. (**a**) Western blot analysis of FAS and ACC in T2DM mice. (**b**) Western blot analysis of ATGL, HSL, and CPT1A in T2DM mice. (**c**) Western blot analysis of p-AMPK and AMPK in T2DM mice. NG, normal diet group; MG, HFD/STZ- induced T2DM; Fen, HFD/STZ-induced T2DM + 250 mg/kg fenofibrate; PC-H, HFD/STZ-induced T2DM + 200 mg/kg proanthocyanidins; PC-L, HFD/STZ-induced T2DM + 50 mg/kg proanthocyanidins. Values are expressed as Mean ± SD, with 9 mice in each group. # *p* < 0.05 and ## *p* < 0.01 compared with the NG group; * *p* < 0.05 and ** *p* < 0.01 compared with the MG group.

**Figure 5 ijms-21-02029-f005:**
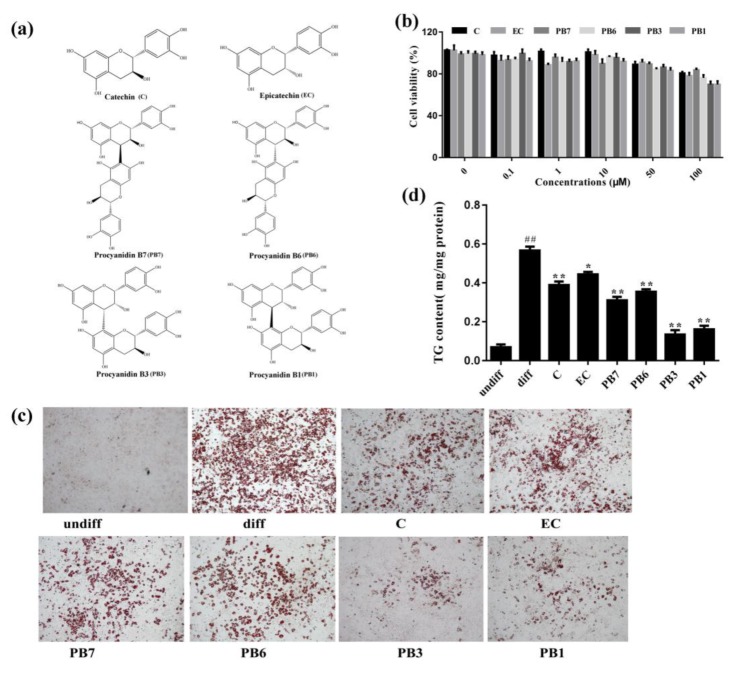
Effects of flavan-3-ols on intracellular lipid accumulation in 3T3-L1 adipocytes. (**a**) Chemical of structures of the six flavan-3-ols purified from proanthocyanidins of *Iris lactea* Pall. var. *Chinensis* (Fisch.) Koidz. (**b**) Cell viability. (**c**) Oil-red-O staining. (**d**) Triglyceride contents. During differentiation, 90% confluent 3T3-L1 preadipocytes were cultured and induced in the Dulbecco’s modified Eagle’s medium (DMEM) medium containing the differentiation cocktail with or without flavan-3-ols for 8 days. undiff, undifferentiated adipocytes; diff, differentiated adipocytes; C, catechin; EC, epicatechin; PB7, procyanidin B7; PB6, procyanidin B6; PB3, procyanidin B3; PB1, procyanidin B1. Values are shown as Mean ± SD of three independent triplicate experiments (*n* = 3). # *p* < 0.05 and ## *p* < 0.01 compared with undifferentiated adipocytes; * *p* < 0.05 and ** *p* < 0.01 compared with the differentiated adipocytes.

**Figure 6 ijms-21-02029-f006:**
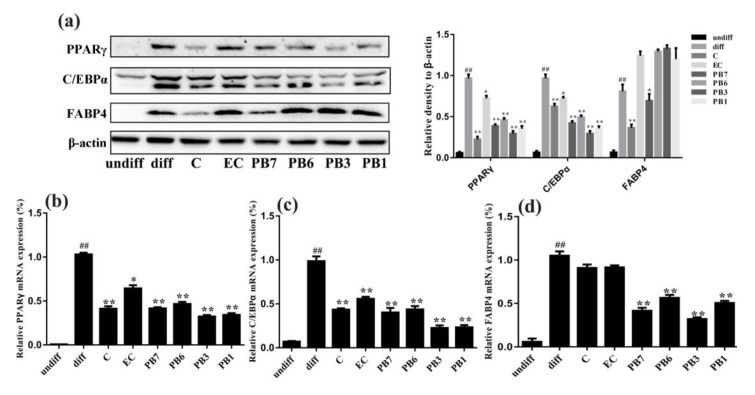
Effects of flavan-3-ols on adipogenesis. (**a**) Western blot analysis of the effects of the six flavan-3-ols on protein levels of PPARγ, C/EBPα and FABP4. RT-PCR analysis of the effects of the six flavan-3-ols on the mRNA levels of (**b**) PPARγ, (**c**) C/EBPα, and (**d**) FABP4. During differentiation, 90% of confluent 3T3-L1 preadipocytes were cultured and induced in the DMEM medium containing the differentiation cocktail with or without flavan-3-ols for 8 days. undiff, undifferentiated adipocytes; diff, differentiated adipocytes; C, catechin; EC, epicatechin; PB7, procyanidin B7; PB6, procyanidin B6; PB3, procyanidin B3; PB1, procyanidin B1. Values are shown as Mean ± SD of three independent triplicate experiments (*n* = 3). # *p* < 0.05 and ## *p* < 0.01 compared with undifferentiated adipocytes; * *p* < 0.05 and ** *p* < 0.01 compared with the differentiated adipocytes.

**Figure 7 ijms-21-02029-f007:**
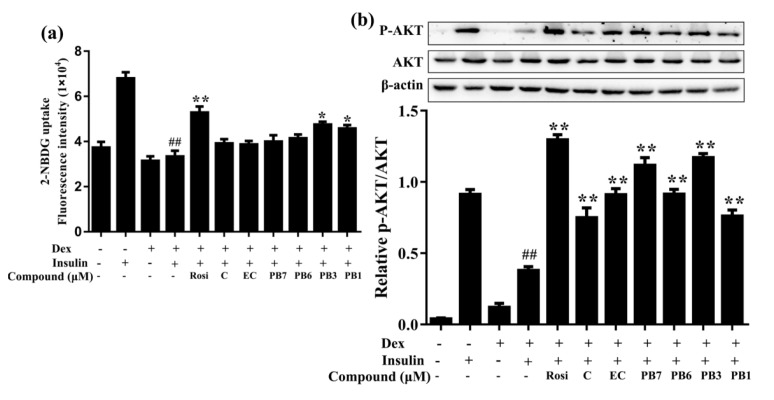
Effects of flavan-3-ols on glucose uptake and phosphorylation of AKT in insulin-resistant 3T3-L1 adipocytes. (**a**) Flavan-3-ols reversed Dex-induced decrease in insulin stimulated glucose uptake in 3T3-L1 adipocytes. (**b**) Western blot analysis of the effects of flavan-3-ols on protein expression of p-AKT. Fully differentiated 3T3-L1 adipocytes were treated with Dex for 48 h in the presence or absence of flavan-3-ols or rosiglitazone. Rosi, rosiglitazone; C, catechin; EC, epicatechin; PB7, procyanidin B7; PB6, procyanidin B6; PB3, procyanidin B3; PB1, procyanidin B1. Values are shown as Mean ± SD of three independent triplicate experiments (*n* = 3). #*p* < 0.05 and ##*p* < 0.01 compared with normal differentiated adipocytes; **p* < 0.05 and ***p* < 0.01 compared with Dex only treated adipocytes.

**Figure 8 ijms-21-02029-f008:**
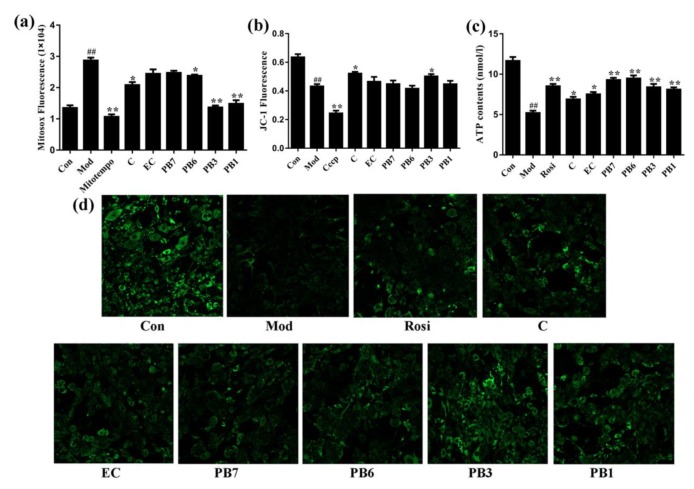
Effects of flavan-3-ols on mitochondrial function in insulin-resistant 3T3-L1 adipocytes. (**a**) Mitochondrial superoxide production. (**b**) Mitochondrial membrane potential. (**c**) The intracellular contents of ATP. (**d**) Mitochondrial mass. Fully differentiated 3T3-L1 adipocytes were treated with Dex for 48 h in the presence or absence of flavan-3-ols or rosiglitazone (100× *g* magnification). Con, normal differentiated 3T3-L1 adipocytes; Mod, Dex alone treated differentiated 3T3-L1 adipocytes; Rosi, rosiglitazone; C, catechin; EC, epicatechin; PB7, procyanidin B7; PB6, procyanidin B6; PB3, procyanidin B3; PB1, procyanidin B1. Values are shown as Mean ± SD of three independent triplicate experiments (*n* = 3). # *p* < 0.05 and ## *p* < 0.01 compared with normal differentiated adipocytes; * *p* < 0.05 and ** *p* < 0.01 compared with Dex only treated adipocytes.

**Figure 9 ijms-21-02029-f009:**
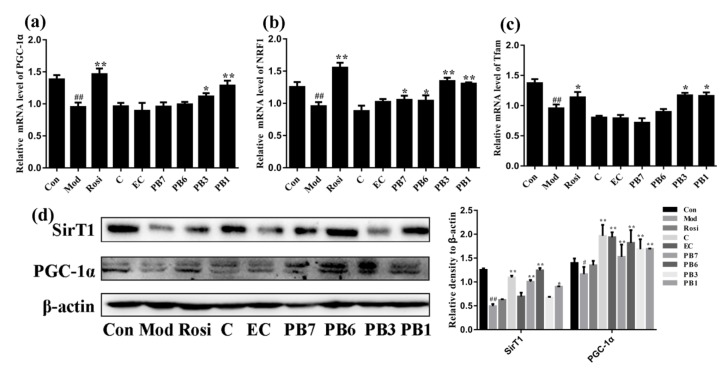
Effects of flavan-3-ols on expression of mitochondrial biogenesis-related protein in insulin-resistant 3T3-L1 adipocytes. RT-PCR analysis of mRNA levels of, (**a**) PGC-1α, (**b**) NRF1, and (**c**) Tfam. (**d**) Western blot analysis of expression of SirT1 and PGC-1α. Fully differentiated 3T3-L1 adipocytes were treated with Dex for 48 h in the presence or absence of flavan-3-ols or rosiglitazone. Con, normal differentiated 3T3-L1 adipocytes; Mod, Dex alone treated differentiated 3T3-L1 adipocytes; Rosi, rosiglitazone; C, catechin; EC, epicatechin; PB7, procyanidin B7; PB6, procyanidin B6; PB3, procyanidin B3; PB1, procyanidin B1. Values are shown as Mean ± SD of three independent triplicate experiments (*n* = 3). # *p* < 0.05 and ## *p* < 0.01 compared with normal differentiated adipocytes; * *p* < 0.05 and ** *p* < 0.01 compared with Dex only treated adipocytes.

**Figure 10 ijms-21-02029-f010:**
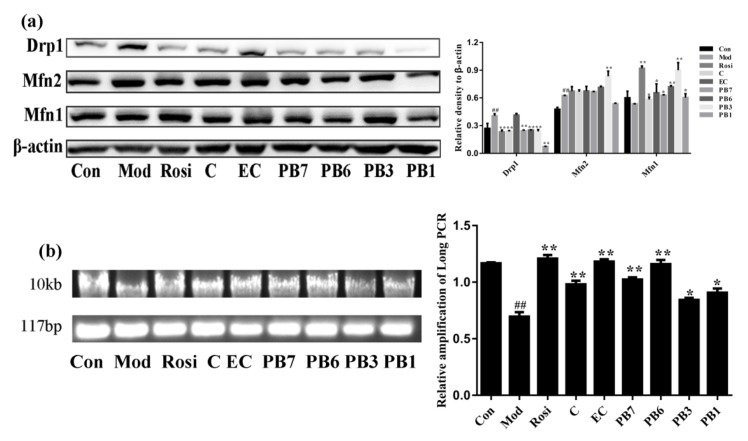
Effects of flavan-3-ols on mitochondrial dynamics and mtDNA damage in insulin-resistant 3T3-L1 adipocytes. (**a**) Western blot analysis of expression of Drp1, Mfn2 and Mfn1. (**b**) Analysis of mtDNA damage based on ratio of long and short DNA fragments after long PCR. Fully differentiated 3T3-L1 adipocytes were treated with Dex for 48 h in the presence or absence of flavan-3-ols or Rosi. Con, normal differentiated 3T3-L1 adipocytes; Mod, Dex alone treated differentiated 3T3-L1 adipocytes; Rosi, rosiglitazone; C, catechin; EC, epicatechin; PB7, procyanidin B7; PB6, procyanidin B6; PB3, procyanidin B3; PB1, procyanidin B1. Values are shown as Mean ± SD of three independent triplicate experiments (*n* = 3). # *p* < 0.05 and ## *p* < 0.01 compared with normal differentiated adipocytes; * *p* < 0.05 and ** *p* < 0.01 compared with Dex only treated adipocytes.

**Figure 11 ijms-21-02029-f011:**
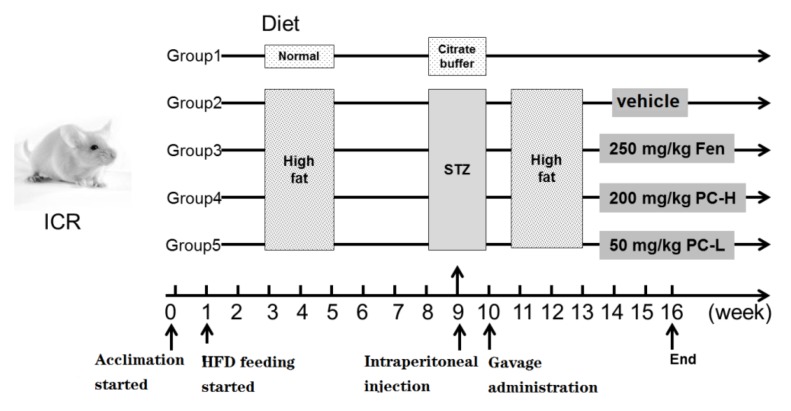
Experimental design of animal study. Group 1, normal control group; Group 2, model group; Group 3, Fen group; Group 4, PC-H group; Group 5, PC-L group.

**Table 1 ijms-21-02029-t001:** Proanthocyanidins Identified by UPLC-Triple-TOF/MS from *Iris lactea* Pall. var. *Chinensis* (Fisch.) Koidz.

Peak No.	Category	tz (min)	M + (*m*/*z*)	MS/MS (*m*/*z*)
1	3,4-Dihydroxybenzoic acid	4.48	153.0218	65/91/109
2	procyanidin B1	6.40	577.1349	125/245/289/407/425/451
3	3’-O-(1-hydroxy-6-oxo-2-cyclohexene-1-carboxylic acid ester) of procyanidin B1	6.63	715.1663	289/419/425/571
4	6-[(1S)-3-methoxy-3-oxo-1-(2,4,5-trihydroxyphenyl)propyl]catechin	6.68	499.1233	245/289/345/389
5	procyanidin B3	6.87	577.1342	125/245/289/407/425/451
6	catechin	7.29	289.0720	109/125/203/245
7	fisetinidol (4α,8)-catechin	7.92	577.1342	125/245/289/407/435
8	epiafzelechin-(4β- > 8)-epicatechin	8.66	561.1394	125/245/289/407/435
9	epicatechin	9.37	289.0722	109/125/203/245
10	procyanidin B7	10.61	577.1345	125/245/289/407/425/451
11	procyanidin A1	11.01	591.1136	125/289/407/465
12	norathyriol	12.35	259.0257	159/191/231
13	proanthocyanidin A1	12.82	575.1186	125/289/407/449
14	procyanidin B6	13.14	577.1185	125/289/407/449
15	(2’S,3’R)-9-(5’,6’-dihydroxy-2’-hydroxymethyl-2’,3’-dihydrobenzo[b]furan-3-yloxy)-6H-dibenzo[b,d]pyran-6-oe	14.93	575.1185	191/258/299/355
16	hopeaphenol	19.91	905.2597	265/358/451/717/811
17	isohopeaphenol	20.49	905.2605	265/359/451/717/811
18	n-butyl pro-lithospermate	21.11	413.1239	145/218/233/367
19	viniferin	21.84	453.1332	197/225/279/345/359
20	vitisin Β	22.55	905.2603	359/451/545/799811
21	vitisin C	23.43	905.2602	359/545/693/799811

**Table 2 ijms-21-02029-t002:** Primers used for RT-PCR and PCR analysis.

Target Gene	Primer Sequence	Size (bp)	Accession Numbers
PPARγ	Forward: 5’-CCTGGCAAAGCATTTCTATG-3’	100	XM_017321456
	Reverse: 5’-TGGTGATTTGTCCGTTGTCT-3’		
C/EBPα	Forward: 5’- CGGCGGGAACGCAACAACAT -3’	109	NM_001287514
	Reverse: 5’- GGCGGTCATTGTCACTGGTC -3’		
FABP4	Forward: 5’-TCACCTGGAAGACAGCTCCT-3’	182	XM_024406
	Reverse: 5’-AATCCCCATTTACGCTGAT-3’		
PGC-1α	Forward: 5’-CGGAAATCATATCCAACCAG-3’	243	XM_006503779
	Reverse: 5’-TGAGGACCGCTAGCAAGTTTG-3’		
NRF1	Forward: 5’-TGGTCCAGAGAGTGCTTGTG-3’	184	NM_001361693
	Reverse: 5’-TTCCTGGGAAGGGAGAAGAT-3’		
Tfam	Forward:5’-GGAATGTGGAGCGTGCTAAAA-3’	118	NM_009360
	Reverse:5’-TGCTGGAAAAACACTTCGGAATA-3’		
β-actin	Forward: 5’-CCTGAGGCTCTTTTCCAGCC-3’	110	NM_007393
	Reverse:5’-TAGAGGTCTTTACGGATGTCAACGT-3’		
Long fragment	Forward: 5’-TACTAGTCCGCGAGCCTTCAAAGC-3’	8636	AJ512208.1
	Reverse: 5’-GGGTGATCTTTGTTTGCGGGT-3’		
Short fragment	Forward: 5’-CCCAGCTACTACCATCATTCAAGT-3’	117	NC_005089
	Reverse: 5’-GATGGTTTGGGAGATTGGTTGATG-3’		

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
