# Peer review of "Proanthocyanidins Ameliorated Deficits of Lipid Metabolism in Type 2 Diabetes Mellitus Via Inhibiting Adipogenesis and Improving Mitochondrial Function"

_ijms, 2020, doi:10.3390/ijms21062029_

Round 1

Reviewer 1 Report

The article “Proanthocyanidins ameliorate deficits of lipid 2 metabolism in type 2 diabetes mellitus via inhibiting 3 adipogenesis and improving mitochondrial function” presented by Tie et al. is well written and organized, and the topic is relevant for the journal. In my opinion, the manuscript is publishable after minor revisions.

I suggest the following:

  • Introduction – row 52: being mentioned in the text for the first time, the name of lactea should be presented as binomial name.
  • Results – row 77: “The calibration curves” should be replaced by “linear regression equation for …”.
  • Results – row 106: being first mentioned in the text, the abbreviation “STZ” must be replaced by „streptozocin (STZ)”.
  • Results – rows 108-116: the codification of mice groups is not properly explained. For example, it is not stated what PC-H and PC-L groups represent. A full description of the groups exists in the chapter “Materials and Methods”, rows 485-498, but there MG is referred as model group, and in the rows mentioned above MG is referred as “high-fat-diet group”. My suggestion is either to describe a little bit more the codification, or to move the “Materials and Methods” chapter before “Results” chapter, making it much easier for the reader to understand the text.
  • Materials and methods – row 498: I suggest the authors to explain in the text why they choose oral gavage for the administration of fenofibrate and proanthocyanidins.
  • Materials and methods – row 481: “top-water” should be replaced with “tap-water”.
  • Conclusions: the results of the study are well summarized in “Conclusions” chapter, but it misses an overall point of view.
  • This manuscript would benefit from a review by a native English speaker to assist with the sentence construction and spelling.

Reviewer 2 Report

The manuscript submitted for review is interesting, with valuable results. However, before considering for publication, a revision is necessary, considering the following aspects:

1. Introduction - Rows 56-57. Reference [12] provided does not support the statement. Please insert relevant works for the actions described.

2. Introduction - Please extend the part presenting potential uses of I. lactea

3. Introduction- Please provide the binomial name of I. lactea at first mention in the introduction.

4. Results - Row 77 - The equation of the calibration curve was.... (not the calibration curve)

5. Materials and methods- Where the seeds properly identified? Please present details.

6. Materials and methods - Why did the authors choose oral gavage for drug administration? Please detail.

7. Materials and methods - According to the IJMS instructions, images of cells and western blots should be large enough to see the relevant features. Please provide those images.

8. Materials and methods - Please provide all the parameters for the SCO2 extraction (pressure, mass flow rate, characteristic particle size if avaialable, etc.)

9. Materials and methods - Row 481 - probably tap-water

10. Conclusions - I do miss a take-home message. Please present the main conclusion of the study in a sentece that could be easily understood by the a more general public.

For the entire manuscript - please carefully check the mansucript for English (see for example the Materials and methods section) and style errors (i.e. - in the Results section please use present tense to reffer to tables or figures, and past tense to describe completed experiments; in the Discussions section - use past tense to summarise findings and present tense to interprete the results and ther significance, etc.).

Please re-number figures and tables, following their number of appearance

Reviewer 3 Report

  1. Data from some of the western blots are not clear. Most of them have to be repeated with appropriate quantifications. For example Figs. 5a (ACC), 5b (HSL), 11a (Drp1).

Some of the merits are: Analysis of serum biochemical markers in diabetic mice models to describe the effects of proanthocyanidins. Lipid accumulation experiments in 3T3-L1 cells are appreciable.

Major comments: Western blots for Fig.5: ACC, ATGL should be repeated. 

Minor comments: Line 26: high-fat-diet/streptozocin (STZ)-induced T2DM in mice.   

STZ-induced diabetes should be in general type 1 diabetes, as it kills pancreatic beta cells, authors may designate STZ-induced diabetes.  

Round 2

Reviewer 3 Report

The authors improved the manuscript with my suggestions.